# Towards Fine-tuning-free Few-shot Classification: A Purely Self-supervised Manner

## Abstract

One of the core problems of supervised few-shot classification is adapting generalized knowledge learned from substantial labeled source data to rarely labeled novel target data. What makes it a challenging problem is how to eliminate undesirable inductive bias introduced by labels when learning generalized knowledge during pre-training or adapting the learned knowledge during fine-tuning. In this paper, we propose a purely self-supervised method to bypass the labeling dilemma, focusing on an extreme scenario where a few-shot feature extractor is learned without fine-tuning. Our approach is built on two key observations from recent advancements in style transfer learning and self-supervised learning:1) high-order statistics of feature maps in deep nets encapsulate distinct information about input samples, and 2) high-quality inputs are not essential for obtaining high-quality representations. Accordingly, we introduce a variant of the vector quantized variational autoencoder (VQ-VAE) that incorporates a novel coloring operation, which conveys statistical information from the encoder to the decoder, modulating the generation process with these distinct statistics. With this design, we find that the statistics derived from the encoder's feature maps possess strong discriminative power, enabling effective classification using simple Euclidean distance metrics. Through extensive experiments on standard few-shot classification benchmark. We show that our fine-tuning-free method achieves competitive performance compared to fine-tuning-based and meta-learning-based approaches.

## 1 Introduction

Just like human beings born with few-shot recognition ability, the large-scale self-supervised pre-training model demonstrates extraordinary "few-shot ability" in computer vision recognition tasks (Radford et al., 2021; Jia et al., 2021; Chen et al., 2023) and natural language understanding tasks (Brown et al., 2020). High-capacity models combined with large-scale training data seem to provide a straightforward solution to few-shot learning. However, the "few-shot" is ill-posed in the context of recent large-scale pre-training paradigms because of the possibility of information leakage between the training and testing stage (Pham et al., 2023). Specifically, it is hard to tell to what extent the few-shot ability comes from a large model's memorization. As the training data scales to hundreds of millions (e.g. 400 million image-text pairs for CLIP (Radford et al., 2021)), the dataset partition of training and testing becomes ambiguous. This ambiguity offers the high-capacity model more opportunities to disguise its memorization as a few-shot recognition ability. Thus, we concentrate on learning a few-shot feature extractor under low-data settings in a self-supervised manner without fine-tuning.

Recently, several works show that a simple supervised pre-trained feature extractor fine-tuned with limited novel data performs well in a few-shot classification task (Chen et al., 2020b; Dhillon et al., 2020). Along with this two-staged approach, self-supervised learning can either be used as an auxiliary task to boost the performance of both stages (Gidaris et al., 2019; Yang et al., 2022; Liu et al., 2021; Su et al., 2020) or be a substitution of the pre-training strategy in the first stage (Poulakakis-Daktylidis & Jamali-Rad, 2024a; Medina et al., 2020; Lu et al., 2022a; Chen et al., 2021a). All of these works alleviate the over-confident inductive bias introduced by labels of source classes with self-supervised learning. Specifically, the latter method called unsupervised few-shot learning(U-FSL) removes the label dependency from source data completely and still demonstrates surprisingly

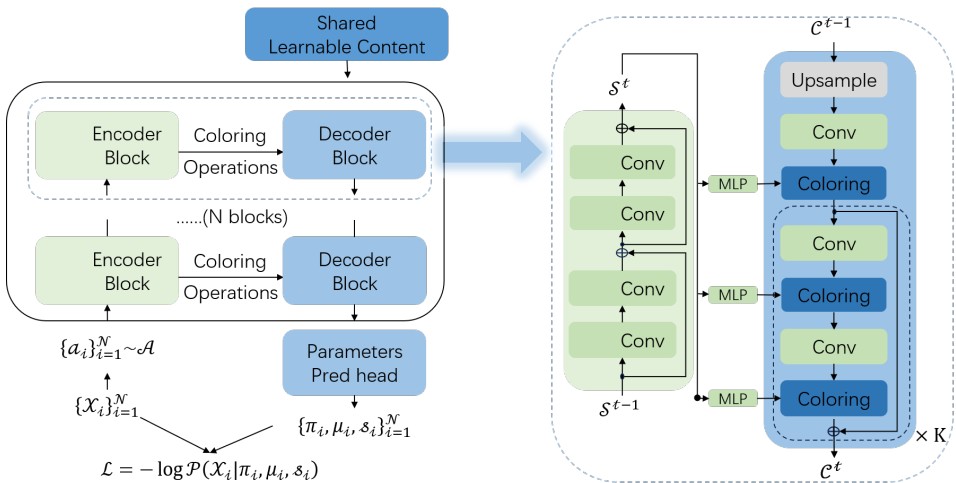

Figure 1: Model overview. As shown on the left side, we use an encoder-decoder architecture for our denoising VQ-VAE. The coloring operations between encoder-decoder pairs make our model different from existing VAE models. On the right side, we detailed our configuration of coloring operations between encoder-decoder pairs. We incorporate vector quantization operation into the coloring operation and display its detailed architecture in Figure 2. We omit weight standardization, normalization, and activation layers for brevity.

few-shot recognition performance with a supervised fine-tuning stage. However, considering the scarcity of data in the fine-tuning stage, the supervised fine-tuning under few-shot settings may lead to other problems (Poulakakis-Daktylidis & Jamali-Rad, 2024a). We follow this line of work on U-FSL and take a pioneering step to remove the label dependency in the fine-tuning stage.

Most U-FSL methods take contrastive learning as the unsupervised pre-training approach in source classes (Poulakakis-Daktylidis & Jamali-Rad, 2024a; Medina et al., 2020; Lu et al., 2022a; Chen et al., 2021a). As we all know, strong data augmentation plays a significant role in contrastive learning (He et al., 2020; Chen et al., 2020a; Caron et al., 2021; Zbontar et al., 2021; Bardes et al., 2022). This is also true for the pre-training stage in U-FSL as observed in (Lu et al., 2022a). Intuitively, strong data augmentation destroys the object semantic information of input images in source datasets. This destruction makes the learned representations less biased towards objects in source classes, and thus can easily transfer to novel target classes with limited-data fine-tuning. Additionally, recent mask-based image models (MIM) (Feichtenhofer et al., 2022; He et al., 2021; Tong et al., 2022; Xie et al., 2022) show that highly masked input combined with self-reconstruction tasks can force the model to learn meaningful representations. Thus, we believe that high-quality images are unnecessary for generalizable representations.

What's more, recent style(domain)-transfer literature (Huang & Belongie, 2017; Li et al., 2017; Ulyanov et al., 2016; Li et al., 2016) show that high-order statistics calculated from feature maps of deep nets contain "style(domain)" information about the input image. We can transport these "style(domain)" information to another image with these statistics. The styleGANs (Zheng et al., 2020; Karras et al., 2020; 2021; 2019) further show that we can perform fine-grained semantic control of a generative process with these statistics. Notably, the semantic control signals(i.e. the statistics) of StyleGANs can be learned directly from white noises. It's reasonable to infer that we can also form these distinct statistics from destructive inputs so that we can use them as discriminative representations. Furthermore, the feature maps are discretized at spatial dimensions when calculating these statistics. t's natural to vector quantize the latent space as (van den Oord et al., 2018; Razavi et al., 2019) when design the architecture.

With all these observations, we propose a denoising VQ-VAE model with statistical conveyers from encoder to decoder as shown in Fig.1. Similar to the styleGANs, the encoder takes as input a corrupted image and provides modulating signals for the decoder and the decoder to reconstruct the original input in pixel space. The corrupted input of the encoder makes it concentrate more on abstract information about the input instead of the object itself or some shortcut attributes(e.g. color,

background). The pixel reconstruction task of the decoder determines what modulating signals are the principal components of the original image. In general, the contributions of this paper are summarized as follows:

- We design a coloring operation that transmits a high-order statistical signal from the encoder to the decoder of a VQ-VAE model. With this design, we empirically show that these high-order statistics benefit both the pre-training and evaluation stages.

- We augment our VQ-VAE with noisy input during pre-training i.e., denoising task. Surprisingly, we find that these noisy-adding procedures can even boost few-shot recognition performance during evaluation without any fine-tuning.

- We empirically demonstrate effectiveness of our method in mini-ImageNet (Vinyals et al., 2017) and show prospects of **fine-tuning-free** U-FSL.

## 2 RELATED WORK

### 2.1 UNSUPERVISED FEW-SHOT LEARNING

Recently, the pre-training method in source classes shifts from supervised learning to unsupervised learning. U-FSL is a promising direction that can advance few-shot learning to a new era of high-capacity models pre-trained with large-scale unlabeled data. Existing research on U-FSL can be roughly divided into two categories:meta-learning approaches (Lee et al., 2020; Ye et al., 2022; Khodadadeh et al., 2019; Jang et al., 2023) and contrastive learning approaches (Lu et al., 2022b; Chen et al., 2021b; Poulakakis-Daktylidis & Jamali-Rad, 2024b). Both of them have an unsupervised pretraining stage in source classes followed by a supervised fine-tuning stage in novel target classes. As demonstrated in (Tian et al., 2020), good representations are significant for few-shot learning. All these works try to learn more generalizable representations in the unsupervised pre-training stage. The former inherits motivation from traditional meta-learning (Finn et al., 2017; Nichol et al., 2018) but collects meta-training episodes in a heuristic manner(e.g. augmentation views (Khodadadeh et al., 2019)) while the latter employs contrastive learning as the unsupervised pre-training strategy. As we have observed, high-order statistics are significant, and high-quality inputs are unnecessary. We replace contrastive learning with a VQ-VAE-based self-reconstruction paradigm, which is consistent with the discretized nature when we calculate the statistics. We take an exploratory step to remove label dependency in the fine-tuning stage by utilizing these high-order statistics as discriminative representations and directly performing nearest-neighbor classification with them just like (Snell et al., 2017).

### 2.2 SEMANTIC DISENTANGLEMENT WITH STATISTICS

Many recent style transfer algorithms show that we can disentangle the style and content of an image by some statistics(e.g. gram matrix (Gatys et al., 2015; Ulyanov et al., 2017), variance (Huang & Belongie, 2017; Dumoulin et al., 2017), covariance (Li et al., 2018c; 2017; Cho et al., 2019)) calculated from feature maps of a pre-trained deep network(e.g. VGG-19). Thus, the style of an image can be transferred by these statistics. Furthermore, several works demonstrate that these statistics not only can disentangle distinct semantics such as style and content but also can disentangle fine-grained semantics(e.g. hair, pose, freckles) in a portrait (Zheng et al., 2020; Karras et al., 2019), categorical semantics of different classes (Siarohin et al., 2019), or domain semantics of different datasets (Li et al., 2016; Chen et al., 2019). More importantly, some of these works show that statistics computed across the deep neural network provide a high-to-low semantic abstraction (Gatys et al., 2015; Karras et al., 2019). This discriminative ability of statistics is also demonstrated in general vision classification task (Li et al., 2018b;a), even in fine-grained classification task (Lin et al., 2017). All of these works show us an intuitive belief that statistics of feature maps in deep neural networks contain distinct information. They can be used to represent discriminative semantics and they are hierarchically distributed across the deep nets. This line of research motivates us in the architectural design of coloring operation and the vector quantization of latent space.

## 3 METHOD

### 3.1 PROBLEM DEFINITION

We follow the commonly used definition of U-FSL in (Lu et al., 2022b; Chen et al., 2021b; Poulakakis-Daktylidis & Jamali-Rad, 2024b). Generally, the U-FSL is divided into two stages: an unsupervised pre-training in source classes(also called base classes) followed by a supervised fine-tuning in disjoint novel classes. In the pre-training stage, all we need are unlabeled data and some augmentations to add noise for our denoising VQ-VAE. We denote the source classes as $D_s = \{x_i\}$, the novel classes as $D_n = \{x_i\}$. Both of them are unlabeled since we do not re-train our model. And $a_i \sim A$ is an augmentation for $x_i$ randomly selected from a set of pre-defined augmentation operations(e.g. random crop, random color jitter, random flip) just like the augmentation strategy used in contrastive learning (He et al., 2020; Chen et al., 2020a). Instead of re-training our model in fine-tuning stage, we directly evaluate our model with some statistics calculated from the pre-trained model using episodes constructed from novel classes. We denote an episode $T = S \cup Q$, where $S$, $Q$ are the support set and query set respectively. $S = \{(x_{nk}, y_{nk})\}_{n=1,k=1}^{N,K}$ is constructed by randomly sampling $N$ classes from novel classes and each class contains $K$ randomly selected samples; $Q = \{x_{nm}\}_{n=1,m=1}^{N,M}$ have $N$ classes same as $S$ and each class contains $M$ randomly selected samples. This is called N-way K-shot in few-shot learning.

### 3.2 DENOISING VQ-VAE FOR U-FSL

In this section, we detailed our pre-training architecture and pre-training strategy for U-FSL. Both the architecture and strategy are fairly simple and almost the same as hierarchical VQ-VAE (Razavi et al., 2019) except that we use high-order statistics as a lateral connection. As shown in the left part of Fig.1, we use a pre-defined random augmentation strategy to add noises to a batch of clean images sampled from source classes $D_s$ so that the content information of objects in $D_s$ are blurred. Then we reconstruct these clean images by a denoising VQ-VAE. Our core innovation lies in the lateral connection between the encoder and decoder pair(the details for one pair of encoder-decoder connections are shown in the right part of Fig.1). For the few-shot classification task, the encoder should not concentrate on content information in source classes as the source classes and novel classes are disjoint. Thus, we leave the contents to the decoder and suppose that the whole source dataset is generated from a content codebook in a latent space like (Razavi et al., 2019). What the encoder does is pass distinct semantic information to modulate the generative process like (Karras et al., 2019). Thus, there should be an information conveyer between the encoder and decoder through which the distinct information from the encoder can be transmitted. As mentioned above, several related works empirically demonstrate that the statistics of feature maps in deep nets can serve as such a tool. We utilize the coloring operation, commonly used in image generation (Cho et al., 2019; Siarohin et al., 2019), as the information conveyer in this work. Suppose $S^{hw \times c}, C^{hw \times c}$ are feature maps from the encoder and decoder respectively. The coloring operation is defined as follows:

$$\hat{C} = \Sigma_s^{\frac{1}{2}}(C - \mu_c) + \mu_s + \gamma(C - \mu_c) + \beta \tag{1}$$

$$\Sigma_s = \frac{1}{hw}(\hat{S} - \mu_s)^\top(\hat{S} - \mu_s) \tag{2}$$

$$\mu_s = \sum_{i=1}^{hw} \hat{S} \qquad \mu_c = \sum_{i=1}^{hw} C \tag{3}$$

where $\hat{S} = MLP(S)$ and $\gamma$ $\beta \in \mathbb{R}^c$ are learnable parameters. Broadcast rules are used where needed. The usage of the coloring operation in our model is shown in Fig.2. We insert this operation right after instance normalization layer (Ulyanov et al., 2017) in decoder such that the statistics(i.e. $\Sigma_s$ $\mu_s$) can compensate the non-contents information that has been whitened out by the instance normalization layer in the generative process.

In practice, we discretize the feature maps across spatial dimensions when we calculate $\Sigma_s$ $\mu_s$ in Eq.2 and Eq.3 for coloring operation. It's natural to discretize the latent space in architectural design. We follow the idea in (van den Oord et al., 2018; Razavi et al., 2019) to learn a discretized codebook using vector quantization. The discretized codebook serves as content information for

the generative process. To avoid the "codebook collapse" problem (Huh et al., 2023; Takida et al., 2022), we use the Gumbel-softmax trick (karpathy, 2021) to sample the codebook for our self-reconstruction task. As we devise the codebook for every decoder block(i.e. a hierarchical manner), we compute logits for the Gumbel-softmax trick in an attention style. Since the query signal is exported from our encoder and the Gumbel-softmax trick is differentiable, there is no need to use straight-through gradient estimation or add any regularization loss to the codebook. The complete vector quantization operation is shown in algorithm 1.

---

**Algorithm 1** Attention-style vector quantization with Gumbel-softmax trick

---

**Input**:feature maps $S \in \mathbb{R}^{hw \times c}$ from encoder;feature maps $C \in \mathbb{R}^{hw \times c}$ from decoder; trainable codebook matrix $T \in \mathbb{R}^{l \times c}$;
**Parameter**: temperature coefficient $\tau$ for Gumbel-softmax
**Output**: vector quantized feature maps $\hat{C} \in \mathbb{R}^{hw \times c}$

1: $Q = MLPs(S + C)$
2: $\hat{T} = GroupWhitening(T)$        ▷ Avoid dimension correlation Huang et al.
3: $K, V = Proj(\hat{T}), Proj(\hat{T})$
4: $\hat{K} = LayerNorm(K)$            ▷ Without scale and shift
5: $\hat{V} = LayerNorm(V)$            ▷ Wihtout scale and shift
6: $logits = matmul(\hat{Q}, \hat{K})$
7: $qmat = gumbel\_softmax(logits, \tau)$
8: $\hat{C} = matmul(qmat, C)$
9: **Return** $\hat{C}$

---

Technically, the architecture of our model is very similar to styleGAN (Karras et al., 2019) except that our encoder takes as input a corrupted version of the target instead of some kind of random noise (e.g. gaussian noise). Since we are not looking for high-quality generators, we also replace the GAN-style loss with a much simpler discretized logistic mixture likelihood on pixels space like (Salimans et al., 2017) as our loss function. Suppose a subpixel value $v$ in an input image, the target of our model can be formulated as:

$$\arg\min_{\theta} -\log P(v|\pi, \mu, s) \tag{4}$$

$$P(v|\pi, \mu, s) = \sum_{l=1}^{L} \pi_i \left[ \sigma\left( \frac{v + 0.5 - \mu_i}{s_i} \right) - \sigma\left( \frac{v - 0.5 - \mu_i}{s_i} \right) \right] \tag{5}$$

where $\theta$ represents the trainable parameters of our model; $\pi$ $\mu$ $s$ are decoded by our decoder and $\sum_{l=1}^{L} \pi_l = 1$ are the mixture indicators. Different from (Salimans et al., 2017)'s implementation, we remove the pixel condition settings due to different contexts.

As mentioned above, statistics in deep nets can serve as the basis for distinguishing different inputs. To evaluate our model, we directly perform nearest-neighbor classification between these statistics of samples in $Q$ and $S$ using Euclidean distance as a metric. The prediction under N-way K-shot settings can be simply formulated as:

$$Prob(s, \hat{s}_i) = \frac{e^{-d(s, \hat{s}_i)}}{\sum_{i=1}^{N} e^{-d(s, \hat{s}_i)}} \tag{6}$$

where $s$ is statistics calculated from feature maps of a query sample; $\hat{s}_i$ is the prototype of statistics for the i-th support set. And $d(\cdot, \cdot)$ is the Euclidean distance function. Notably, we do not perform any fine-tuning during evaluation. Instead, we calculate statistics from the pre-trained model directly.

## 4 EXPERIMENTS

### 4.1 DETAILS OF ARCHITECTURE AND DATASETS

**Datasets:** The commonly used few-shot dataset miniImageNet (Vinyals et al., 2017) is employed to demonstrate the effectiveness of our method. This dataset is constructed from subsets of ImageNet. It contains 100 classes with exactly 600 images in each class. We follow the previous work (Ren et al., 2018) to randomly select 64, 16, and 20 classes for training, validation, and testing, respectively. For training data, we first resize all images to a resolution of $448 \times 448$. Then we follow the practice in constructive learning to add noise with predefined random augmentations(e.g. random crop, color jitter). The noised images are resized to the resolution of $256 \times 256$ for resnet-18, and $128 \times 128$ for Conv4 since it is designed for extremely low-resolution input. To ease the computational burden, the reconstruction resolution is half of the input resolution.

**Architecture:** Our VQ-VAE adopts encoder-decoder architecture as its framework. We use resnet-18 (He et al., 2015) or Conv4 (Vinyals et al., 2017) as encoder backbones for different experimentations. We make one modification for our encoder backbones. To remove inter-sample correlation, we replace batch normalization (Ioffe & Szegedy, 2015) in our encoder backbones with group normalization(Wu & He, 2018) and weight standardization (Qiao et al., 2020) just like (Richemond et al., 2020). The architecture of our decoder is very similar to styleGAN except that the convolution layers in the decoder are wrapped up by weight standardization. We use the nearest in-

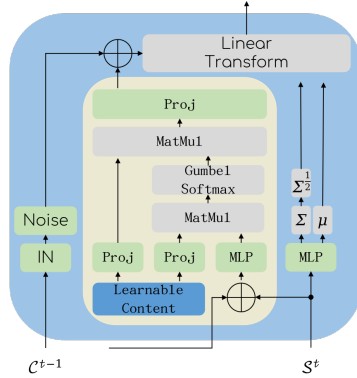

Figure 2: The coloring operation is a linear transformation formulated by Eq.1. This transformation is implemented with components on both sides above. We incorporate an optional attention-style vector quantization(circled by the light yellow rounded rectangle) into this transformation for our VQ-VAE. More information on this vector quantization is detailed in algorithm 1

terpolation followed by a 3x3 convolution layer for upsampling; After that several residual blocks are followed to construct an upsampling block for the decoder. In every decoder block, a coloring operation and an activation layer are employed in sequence after each convolution layer. The discretized codebook for VQ-VAE is incorporated into our coloring operations as shown in Fig. 2. The general structural diagrams are shown in the right part of Fig.1. Apart from the difference in number of blocks, the decoder structure is identical for both resnet-18 and Conv4.

**Other settings:** We export four lateral connections of coloring operation for both resnet-18 and Conv4. The export points are located at the end of the last four stages of resnet-18 and the end of the pooling operation of Conv4. During evaluation, we extract feature maps from these export points for statistics estimation. We insert a MLP block for every lateral connection of Conv4 so that its generative process is as similar as possible to that of resnet-18. We use an iterative method as (Li et al., 2018a;b) for matrix square root in Eq.1.

## 4.2 HIGH-ORDER STATISTICS MATTER

As discussed above, statistics in deep nets play a significant role in distinct information representation. In this section, we empirically demonstrate that these statistics do benefit both the training and testing stages of our VQ-VAE in the context of few-shot recognition. To show the benefits of high-order statistics in the pre-training stage, we instantiate two versions of coloring operation according to Eq.1: 1) full version; 2) mean only. As shown in Fig.3a, the model trained without $\Sigma$ converges more slowly and gets stuck at a suboptimal state during pre-training. Since the $\Sigma$ in Eq.1 gives a linear combination across channel dimension, we take a further look into the eigenvalues of $\Sigma$. As shown in Fig.4, we find that eigenvalues of $\Sigma$ are more divergent, and some of them even tend to zero when td with mean only. This means that $\Sigma$ in Eq.1 can help the model distribute information across channels more stably and evenly, making the model converge faster and better. This better convergence is also shown in Fig.3b, The "full" model outperforms the "mean only" in all representation forms. It is understandable that the mean and covariance perform well in the "mean only" and "full model" respectively. Interestingly, the content of the "full model" outperforms that of "mean only" by a large margin. This is consistent with belief shown in Fig.4 that $\Sigma$ can help the model distribute discriminative information across the nets.

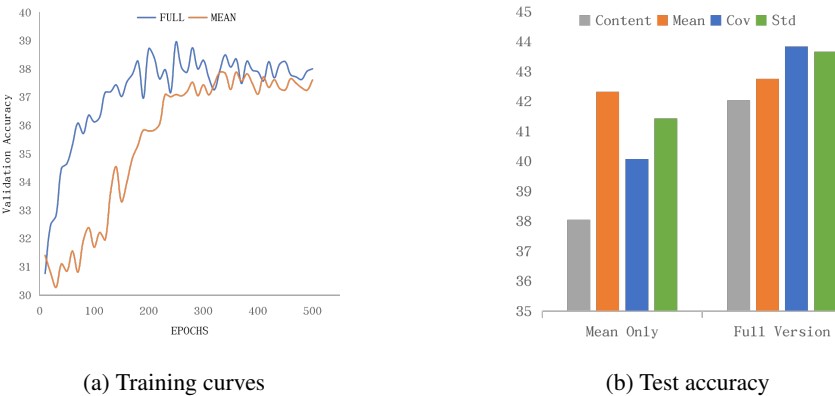

(a) Training curves             (b) Test accuracy

Figure 3: We training 2 versions of Conv4 nets with mini-ImageNet for 800 epochs. We plot the training curves for first 500 epochs with accuracy of content as indicator. On the right side, we plot average few-shot recognition accuracy over 2000 test episodes for different representations

Since we do not retrain our model during the testing stage, the straightforward way to show the benefits of high-order statistics in this stage is to use them as the discriminative basis for the few-shot recognition task directly. Thus, we first pre-train our modified resnet-18 and Conv4 with mini-ImageNet and then perform few-shot classification using feature maps(we term it as "content") and the corresponding statistics calculated from them. As shown in Fig. 5, high-order statistics perform best for different backbones(Fig.5a) and different stages of the same backbone(Fig.5b). Interestingly, the best-performing high-order statistic is covariance for Conv4 and standard deviation for resnet-18. We believe this is due to insufficient samples for estimating covariance in the last stage of resnet-18. As shown in Fig. 5b, since there are enough samples for Conv4 to estimate covariance stably, covariance is consistently better than standard deviation across all stages. All in all, high-order statistics contain more discriminative information for few-shot recognition. This fine-tuning-free superior discriminative power of high-order statistics during evaluation also gives a supplementary explanation of why we shouldn't remove $\Sigma$ in Eq.1 during pre-training.

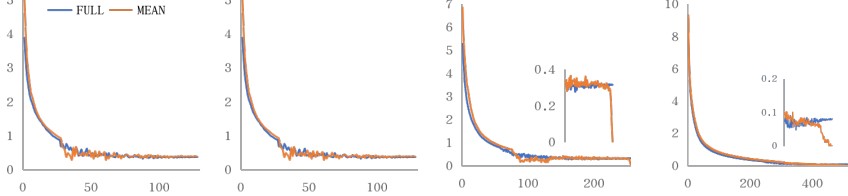

Figure 4: We randomly select a test episode and calculate the corresponding covariance of feature maps at all 4 export points of the 2 pre-trained Conv4 nets. We calculate covariance for each sample as Eq.2 and decompose it to get their eigenvalues. Then, we calculate the mean of the eigenvalues over all samples in this test episode. From left to right, there are plots of export points 1-4 respectively. The horizontal axis represents "dimension" and the vertical axis represents "eigenvalues". We zoom in last several dimensions of Ex-Point3 and Ex-Point4 for a better view.

### 4.3 POST-PROCESSING MATTERS

As mentioned above 1) high-quality inputs are not necessary for high-quality representations; 2) the feature maps are vector quantized in the latent space; 3) statistics are better discriminative representations for few-shot recognition tasks. Accordingly, it's reasonable to provide sufficient samples in the latent space for stable statistics estimation so that better performance can be achieved. Surprisingly, simple augmentations and resolution extension in pixel space work well. The augmentations used can be found in Appendix A.1. To demonstrate the effectiveness of these strategies, we first resize every image in a test episode to a specific resolution and then the resized images are augmented

|              | $128 \times 128$ | $160 \times 160$ | $192 \times 192$ | $224 \times 224$ | $256 \times 256$ |
|--------------|------------------|------------------|------------------|------------------|------------------|
| w/o augs(cov) | $39.96 \pm 0.141$ | $41.12 \pm 0.139$ | $41.14 \pm 0.144$ | $42.55 \pm 0.147$ | $41.86 \pm 0.147$ |
| w/ augs(cov)  | $45.45 \pm 0.153$ | $46.41 \pm 0.153$ | $\mathbf{46.51 \pm 0.151}$ | $46.32 \pm 0.149$ | $46.38 \pm 0.145$ |
| w/o augs(std) | $41.03 \pm 0.153$ | $42.12 \pm 0.144$ | $41.63 \pm 0.147$ | $43.09 \pm 0.143$ | $42.37 \pm 0.153$ |
| w/ augs(std)  | $44.15 \pm 0.150$ | $44.88 \pm 0.155$ | $45.26 \pm 0.150$ | $\mathbf{45.40 \pm 0.155}$ | $45.32 \pm 0.152$ |

Table 1: We train a Conv4 model with mini-ImageNet for 800 epochs. We adopt the conventional settings that report accuracy in ($\% \pm std$) over 2000 test episodes, each with $M = 15$ query shots per class. The worst accuracy is underlined while the best is in bold. The resolution of input image is $128 \times 128$

with pre-defined augmentations to produce several augmented views. After that, both the clean image and those augmented views are fed to our encoder to get augmented representations. Finally, The statistics for one sample are calculated from all those augmented representations in the latent space.

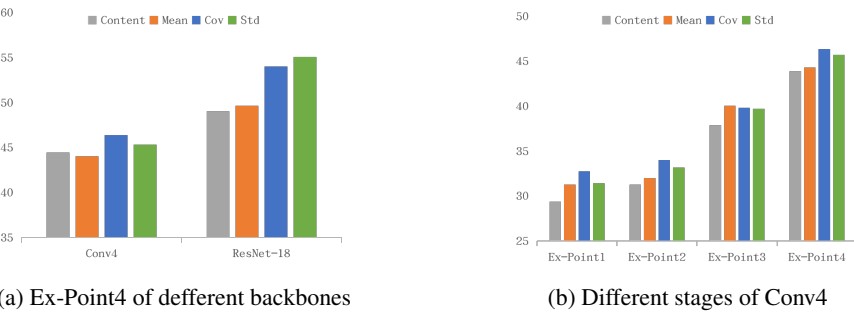

(a) Ex-Point4 of defferent backbones

(b) Different stages of Conv4

Figure 5: Test accuracy of resnet-18 and Conv4. We calculate classification accuracy with different statistics according to Eq.6 and report average accuracy over 2000 test episodes, each with $M = 15$ query shots per class.

As shown in Table 1, The best performance is improved by about 6 percent for covariance and 4 percent for standard deviation when post-processing is used properly. This is reasonable since both augmentations and resolution extension increase samples in latent space so that a better-estimated covariance can be obtained. We plot the eigenvalues in Fig. 6, the rank of covariance is improved by post-processing. These improved ranks provide extra discriminative information for better recognition. Another interesting finding in Table1 is that augmentation is more efficient than resolution extension. We believe this is due to group normalization used in our encoder, and better normalization operations deserve further study. More disscusion can be found in Appendix A.2.

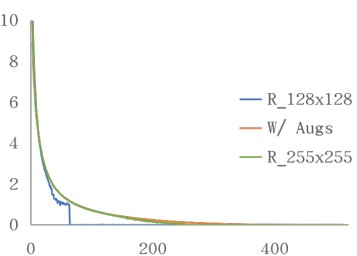

Figure 6: Eigen values of covariance with/without post-processing

### 4.4 COMPARISON WITH BASELINES

In this section, we compare our method with several two-staged baselines including fine-tuning-based and meta-learning-based methods. First of all, it is worth noting that our method not only pre-training in a purely unsupervised manner but also no fine-tuning done during evaluation. In one word, our method is a one-staged U-FSL. From this point of view, our method is competitive when compared with those fine-tuning methods on U-FSL shown in Table 2. This slightly lagging behind in performance is reasonable since our denoising self-reconstruction task relies on high-capacity architecture. As we increase the model capacity, our method outperforms all those representative methods in FSL shown in 3. Whether it is locally supervised fine-tuning (Chen et al., 2020b) or overall fast adaptation (Finn et al., 2017), or global supervised fine-tuning (Snell et al., 2017), our method demonstrates its superiority with simple post-processing. What's more, our method shows a very low variance in interval estimation. With all these comparisons, our method shows that

| Method | backbone | settings | 5-way-1-shot |
|--------|----------|----------|--------------|
| C3LR Shirekar & Jamali-Rad | conv4 | un-pt+sup-ft | $47.92 \pm 1.20$ |
| ProtoTransfer Medina et al. | conv4 | un-pt+sup-ft | $45.67 \pm 0.97$ |
| Meta-GMVAE Lee et al. | conv4 | un-pt+sup-ft | $42.82 \pm 0.45$ |
| PsCo Khodadadeh et al. | conv5 | un-pt+sup-ft | $46.70 \pm 0.42$ |
| Ours(std+augs+256) | conv4 | un-pt+no-ft | $45.32 \pm 0.15$ |
| Ours(cov+augs+256) | conv4 | un-pt+no-ft | $46.38 \pm 0.15$ |

Table 2: Comparisons on mini-ImageNet with Conv4 and unsupervised pretraining strategy. We use a resolution of 256x256 for the input images. The augmentations used are detailed in the Appendix. We adopt the conventional settings that report accuracy accuracies in ($\% \pm std$) over 2000 test episodes, each with $M = 15$ query shots per class. **sup-ft** means supervised fine-tuning, **un-pt** means unsupervised pre-training, **no-ft** means no fine-tuning is used.

good representations with extremely simple post-processing may be sufficient for few-shot learning. These representations can be obtained from the source data in an unsupervised manner. These findings are consistent with (Tian et al., 2020; Raghu et al., 2020). And we take a further step to show that high-order statistics with simple post-processing is a better option than fine-tuning the pre-trained network. Notably, we do not intend to propose a method with SOTA performance, but to show the possibility of fine-tuning-free U-FSL.

| Method | backbone | settings | 5-way-1-shot |
|--------|----------|----------|--------------|
| MatchingNet[†] | resnet-18 | sup-pt+no-ft | $52.91 \pm 0.88$ |
| ProtoNet[†] | resnet-18 | sup-pt+sup-ft | $54.16 \pm 0.82$ |
| Baseline[†] | resnet-18 | sup-pt+sup-ft | $51.75 \pm 0.80$ |
| Baseline++[†] | resnet-18 | sup-pt+sup-ft | $51.87 \pm 0.77$ |
| RelationNet[†] | resnet-18 | sup-pt+sup-ft | $52.48 \pm 0.86$ |
| MAML[†] | resnet-18 | sup-pt+sup-ft | $49.61 \pm 0.92$ |
| Ours(cov+augs+512) | resnet-18 | un-pt+no-ft | $54.25 \pm 0.16$ |
| Ours(std+augs+512) | resnet-18 | un-pt+no-ft | $55.43 \pm 0.16$ |

Table 3: Comparisons with several baselines on mini-ImageNet with resnet-18. The evaluation settings are almost the same with Conv4 except that the input image resolution is 512x512. **sup-pt** means supervised pre-training. Data marked with "†" are borrowed from (Chen et al., 2020b) which are improved versions of original methods.

# 5 CONCLUSION

Recently, there have been many works demonstrating the effectiveness of two-staged few-shot learners including both supervised and unsupervised pre-training methods. In this paper, we propose a new method that directly uses high-order statistics calculated from pre-trained deep nets as discriminative representations so that we do not need any fine-tuning stage. We first design a denoising VQ-VAE and augment it with coloring operations such that high-order statistics residing in the deep nets are discriminative. Then we find that simple post-processing can boost the few-shot recognition performance with these high-order statistics. We also provide some empirical insight into how high-order statistics benefit the training end evaluation of deep nets. Our method has unique advantages in simplicity and adaptability to larger-scale unsupervised pre-training. In summary, we have taken an exploratory step towards fine-tuning-free few-shot learning in a purely unsupervised manner. We hope our research can shed new light on one-staged unsupervised few-shot learning.

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

## A APPENDIX

### A.1 IMAGE AUGMENTATION

The noisy-adding procedure for pre-training is basically inherited from contrastive learning(He et al., 2020; Chen et al., 2020a) except that we add two random channel shuffle and random solarize operations to it. We list them in Table4 in the order in which they are used. Where $p$ means the augmentation proportion in a batch. During evaluation, we use one determined augmentation operation to get one augmented view. There are 11 augmentations for evaluation, which means there are 12 views for one sample during evaluation. We list all the augmentations used during in Table 5. **Interestingly, even though some augmentations are not used in pre-training, they can improve few-shot recognition performance during evaluation**.

| random_crop_resize(size=size,area=$(0.2, 1.0)$) |
| random_solarize(p=0.2) |
| random_channel_shuffle(p=0.2) |
| random_rgb_to_grayscale(p=0.2) |
| random_gaussian_blur(p=0.5) |
| random_flip_left_righ(p=0.5) |
| random_flip_up_down(p=0.5) |

Table 4: Random augmentations in pre-training

| Identity() | invet() |
| rgb_to_gray_scale() | autocontrast() |
| color_jitter() | posterize(bit=4) |
| gaussian_blur(sigma=1.0) | equalize() |
| solarize() | sharpness(factor=0.5) |
| channel_shuffle() | gaussian_noise(stddev=0.1) |

Table 5: augmentations in post-processing

### A.2 DISSCUSION ON POST-PROCESSING

In Table 6, we list the test accuracy of resnet-18 with different configurations of post-processing. The trends presented in this table are basically the same as those in Table 1, except that the standard deviation is the best representation of the few-shot recognition of resnet-18. More notably, even when we use post-processing to increase the number of samples in the latent space, the standard deviation representations consistently outperform the covariance representation by a large margin. We speculate that the covariance estimation of resnet-18 is not as stable as conv4 due to an insufficient number of latent space samples during training. As a result, resnet-18 cannot effectively utilize the sample increment provided by post-processing during evaluation like Conv4. **Thus, improving the stability of covariance estimation during pre-training deserves further study.**

|                | $256 \times 256$ | $320 \times 320$ | $384 \times 384$ | $448 \times 448$ | $512 \times 512$ | $576 \times 576$ |
|----------------|------------------|------------------|------------------|------------------|------------------|------------------|
| w/o augs(cov)  | 46.20            | 48.46            | 50.51            | 51.51            | 52.38            | 52.72            |
| w/ augs(cov)   | 48.70            | 51.63            | 53.15            | 53.63            | 54.04            | **54.25**        |
| w/o augs(std)  | 50.51            | 52.54            | 53.34            | 54.07            | 54.70            | 54.55            |
| w/ augs(std)   | 50.82            | 53.81            | 54.65            | 54.93            | 55.10            | **55.43**        |

Table 6: The resnet-18 is trained with ImageNet for 1600 epochs. We report average accuracy over 2000 test episodes. The resolution of the input image is $256 \times 256$

In Table 7, We give a preliminary exploration of the impact of the normalization layer on post-processing. When we replace group normalization with layer normalization (Ba et al., 2016), the

sensitivity of the covariance representation to resolution extension increases, but the sensitivity to augmentation decreases significantly, and the overall few-shot recognition performance also decreases.

| | $128 \times 128$ | $160 \times 160$ | $192 \times 192$ | $224 \times 224$ | $256 \times 256$ |
|---|---|---|---|---|---|
| ln w/o augs(cov) | 38.69 | 40.24 | 42.19 | 43.04 | 42.85 |
| ln w/ augs(cov) | 43.18 | 44.35 | 44.71 | **45.13** | 44.82 |
| gn w/o augs(cov) | 39.96 | 41.12 | 41.14 | 42.55 | 41.86 |
| gn w/ augs(cov) | 45.45 | 46.41 | **46.51** | 46.32 | 46.38 |

Table 7: The training and testing processes are exactly the same as in Table 1 except that we replace the group normalization in the Conv4 encoder with layer normalization.

