# OpenReview forum: "Towards Fine-tuning-free Few-shot Classification: A Purely Self-supervised Manner"
_ICLR.cc/2025/Conference — Submitted to ICLR 2025_

### Official Review · Reviewer_iijw · 2024-10-17

**Soundness:** 2
**Presentation:** 1
**Contribution:** 2
**Rating:** 3
**Confidence:** 3

**Summary:**

The paper addresses the challenge of adapting generalized knowledge to novel tasks with minimal labeled data, specifically in few-shot classification. The authors propose a method to bypass traditional fine-tuning by leveraging a purely self-supervised approach. Their key innovation is a modified VQ-VAE that incorporates a "coloring operation" to transmit high-order statistics between the encoder and decoder. This allows for effective classification using simple Euclidean distance metrics, without the need for high-quality inputs or large-scale fine-tuning.

**Strengths:**

* Introduction of a coloring operation that conveys high-order statistics in a VQ-VAE, improving both pre-training and evaluation.

* Empirical demonstration of the effectiveness of this approach on the mini-ImageNet dataset, showing that the method achieves competitive performance compared to fine-tuning-based approaches while eliminating the need for fine-tuning.

**Weaknesses:**

* The experimental setup is overly simplistic. Validation was only conducted on miniImageNet, and the image resolution was increased to 256, which is inconsistent with most methods. Additionally, the compared methods are outdated, such as Baseline++ and MAML, which were proposed several years ago.

* The so-called "fine-tuning-free" approach is essentially still based on metric learning with distance comparisons, similar to ProtoNet.

* The writing of the paper is difficult to understand, and the method is not clearly explained. For instance, after computing $V$ in Algorithm 1, it is not utilized in the subsequent steps.

**Questions:**

* The authors claim that high-quality inputs are not necessary for obtaining useful representations. Can the authors further clarify how robust is the method to extreme noise or data corruption?

---

### Official Review · Reviewer_H2Vt · 2024-11-01

**Soundness:** 2
**Presentation:** 3
**Contribution:** 2
**Rating:** 5
**Confidence:** 4

**Summary:**

This paper introduces a fine-tuning-free approach to few-shot classification using a purely self-supervised method. The authors propose a variant of VQ-VAE that incorporates a coloring operation to transmit high-order statistical information from the encoder to the decoder. This method enables the model to perform classification using Euclidean distance on the derived feature map statistics without fine-tuning. Through experiments on standard few-shot classification benchmarks, the method demonstrates competitive performance compared to traditional fine-tuning-based and meta-learning approaches.

**Strengths:**

- The paper is easy to follow.
- The paper presents an innovative approach to few-shot classification by removing the need for fine-tuning, focusing on purely self-supervised learning, which is less commonly explored.
- The approach uses high-order statistics for representation, which is a novel method for reducing the reliance on fine-tuning while retaining discriminative information.
- Despite being fine-tuning-free, the proposed method achieves results comparable to traditional few-shot learning methods, suggesting potential for practical application in scenarios where fine-tuning is infeasible.

**Weaknesses:**

- The paper focuses primarily on mini-ImageNet, which, while common, limits insights into how this method generalizes across more diverse datasets or tasks, e.g. tiered-ImageNet, Meta-dataset
- Although high-order statistics provide significant benefits, the approach’s reliance on them might limit performance in tasks where these statistics are insufficient for capturing fine-grained details.
- The VQ-VAE with coloring operations introduces additional complexity compared to standard architectures, which might hinder real-time or resource-constrained applications.
- Some details, such as specific parameter settings or detailed statistical analyses, are sparsely provided, making it harder for readers to replicate or fully assess the robustness of the approach.

**Questions:**

The paper could benefit from more extensive comparisons with the latest few-shot learning approaches to better contextualize its performance.

---

### Official Review · Reviewer_z6Uv · 2024-11-03

**Soundness:** 2
**Presentation:** 2
**Contribution:** 2
**Rating:** 5
**Confidence:** 4

**Summary:**

This paper studies the fine-tuning free few shot learning in a self-supervised learning manner. It is build on two observations, one for the high-order statistics and another for the quality of representaions. These issues are addressed by incorporating the coloring operation to the VQ-VAE,conveying the statistical information from the encoder to thedecoder. Experiments show the proposed self-supervised learning outperfomrs the existing FSL paradigm.

**Strengths:**

1-This paper is easy to follow.

2. The experiment results are good.

3-It seems the proposed fine -tuning manner can reduce the computational complexity of FSL.

**Weaknesses:**

1-The experiments are not enough, It is suggested to compare to more sota methods of vq-vae.

2-VQ-VAE is used in this paper, I guess it will increase the computational resources.

3-I didn't understand what is the high order statics of feature map? How to measure the statics?

4-Experiments on other dataset，CUB，tiered-imagenet etc，and on other backbones are also suggested.

5-Why use the VQ-VAE in this work, other self-supervised method, like the contrastive learning, can also be used. Why not use the contrastive learning scheme?

**Questions:**

See above

---

### Official Review · Reviewer_WiRn · 2024-11-03

**Soundness:** 3
**Presentation:** 3
**Contribution:** 2
**Rating:** 5
**Confidence:** 4

**Summary:**

The paper proposes a fine-tuning free unsupervised few-shot learning approach that is inspired by learning features from higher order statistics of the feature map supported with a generative based coloring pre-text task. In order to learn the backbone that act as a feature extractor, the paper uses a VQ-VAE model along with a coloring operation that transmits a high-order statistical signal from the encoder to the decoder using the excess  examples from the base classes. The few shot classification on novel classes is performed without a fine-tuning stage and uses nearest-neighbor classification between the query samples and support samples of the novel classes. The paper proposes an novel unsupervised approach where label values of the source and target domain are not used training the feature extractor.

**Strengths:**

The paper explores a generative based few shot learning approach that uses coloring operation supported by higher order feature map statistics to learn the backbone of the model. The proposed framework demonstrates competitive performance on mini-imagenet dataset with various existing baselines. The paper provides detailed ablation experiments to analyze/evaluate the various higher order statistics and input image resolution that could be used during the training/inference stage of the models (ie mean, std, covariance). The paper is easy to follow and has pseudocode that explains the methodology.

**Weaknesses:**

As mentioned in the paper, the proposed approach is a fine-tuning free approach, it would be helpful for a reader to get a better understanding of the following:

1. Comparison with SOTA baselines that also use unsupervised pre-training strategies on the base classes, like BECLR and UniSiam. Also a discussion about the performance of just using unsupervised pre-training baseline with nearest neighbor classification could help the readers understand the contribution of the proposed approach and this could also act as a good baseline for the proposed approach (As it would become a fine-tuning free Unsupervised FSL appraoch)


2. The performance boost doesn’t look substantial on 1-shot setting, also 5-shot evaluation seems to be missing that could help the readers evaluate the efficacy of the approach.


3. An ablation on the backbone of the model used (resnet18, conv4), how does this translate to the proposed VQVAE architecture. What are the number of parameters used in the VQVAE approach compared to other baseline, are the number comparable? Keeping the comparable number of parameters for the backbone and same input image resolution size could provide a clearer picture to the readers.


Some minor typos/suggestion:
1. Line 102 : “t’s natural” -> its natural
2. Reference for Chen et al., 2021b and Poulakakis-Daktylidis & Jamali-Rad, 2024b are repeated
3. Line 278: “constructive” -> contrastive
4. Line 318: “when td with mean only”

**Questions:**

It would be helpful if the paper can answer/comment the following questions/suggestions:

1. It would be great to see a discussion/some sample images from the VQ-VAE model in order to see the performance of this generative model.

2. How does the performance change in 5-shot setting and other backbone extractors (like wide-resnet or higher capacity resnet) compared to the baseline approaches.

---

### Meta-Review · Area_Chair_CFbq · 2024-12-15

**Metareview:**

In this paper, the authors proposed a fine-tuning-free few-shot learning approach using a variant of VQ-VAE that leverages high-order statistics for classification without labeled data. While the method demonstrates competitive performance on mini-ImageNet and reduces computational overhead, it lacks evaluations on diverse datasets (e.g., tiered-ImageNet) and comparisons with recent methods like BECLR. Reviewers raised issues such as ambiguities in the methodology, limited dataset scope, reliance on older baselines, and the added complexity of the VQ-VAE architecture. The authors decided not to provide responses during the rebuttal phase. Therefore, the final decision is rejection.

**Additional Comments On Reviewer Discussion:**

The authors didn't provide responses during the rebuttal period. There were no discussions between authors and reviewers.

---

### Decision · Program_Chairs · 2025-01-22

Reject